# Self-Reported Sedentary Behavior and Metabolic Syndrome among Children Aged 6–14 Years in Beijing, China

**DOI:** 10.3390/nu14091869

**Published:** 2022-04-29

**Authors:** Ning Yin, Xiaohui Yu, Fei Wang, Yingjie Yu, Jing Wen, Dandan Guo, Yuanzhi Jian, Hong Li, Liyu Huang, Junbo Wang, Yao Zhao

**Affiliations:** 1Department of Nutrition and Food Hygiene, School of Public Health, Peking University, Beijing 100191, China; yinning@bjmu.edu.cn (N.Y.); wangfei.duzhuo@163.com (F.W.); wenjing@bjmu.edu.cn (J.W.); 1610306107@pku.edu.cn (Y.J.); 2Beijing Center for Disease Prevention and Control, Beijing 100013, China; yxh770770@sina.com (X.Y.); yyj.jane.1982@163.com (Y.Y.); aldblm.tm@163.com (D.G.); benlih@sina.com (H.L.); huangly2006@163.com (L.H.); 3Beijing Key Laboratory of Toxicological Research and Risk Assessment for Food Safety, Peking University, Beijing 100191, China

**Keywords:** sedentary behavior, metabolic syndrome, children

## Abstract

(1) Objective: This study aimed to examine the prevalence of metabolic syndrome (MetS) in children aged 6–14 years in Beijing, and to determine whether sedentary behavior is a risk factor. (2) Methods: Using a multistage stratified cluster random sampling method, 3460 students were selected for the Nutrition and Health Surveillance in Schoolchildren of Beijing (NHSSB). Data on children’s sedentary behavior time and MetS indicators were collected using the questionnaires, physical measurements, and laboratory tests. MetS was defined according to the CHN2012 criteria, and logistic regression analysis was used to compare the effects of different sedentary time on MetS and its components. (3) Results: The overall prevalence of MetS among children aged 6–14 in Beijing was 2.4%, and boys, suburban children, and older age were associated with a higher prevalence (χ^2^ values were 3.947, 9.982, and 27.463, respectively; *p* < 0.05). In boys, the prevalence rates of abdominal obesity, hyperglycemia, high triglycerides (TG), and low high-density lipoprotein cholesterol (HDL-C) were higher in the high-level sedentary behavior group than those in the low-level sedentary behavior group (*p* < 0.05); and in girls, the prevalence rates of high TG, low HDL-C, and MetS were higher in the high-level sedentary behavior group than those in the low-level sedentary behavior group (*p* < 0.05). After adjusting for confounding factors, the multivariate logistic regression results showed that compared with children with low-level sedentary behavior, the risks of abdominal obesity and low HDL-C were higher in boys with high-level sedentary behavior (odds ratio (OR) 1.51, 95% confidence interval (CI) 1.10–2.07, *p* = 0.011; OR 2.25, 95% CI 1.06–4.76, *p* = 0.034, respectively); while the risk of abdominal obesity was higher in girls with medium and high-level sedentary behavior (OR 1.52, 95% CI 1.01–2.27, *p* = 0.043; OR 1.59, 95% CI 1.04–2.43, *p* = 0.032, respectively). (4) Conclusions: Higher sedentary behavior time was related to the higher risk of MetS components among children aged 6–14 in Beijing. Reducing sedentary behavior may be an important method for preventing metabolic diseases.

## 1. Introduction

Metabolic syndrome (MetS) is a group of clinical syndromes that are closely related to lifestyle and characterized by a combination of obesity, hyperglycemia, hypertension, and dyslipidemia, including elevated triglyceride (TG) and low high-density lipoprotein cholesterol (HDL-C) [1,2].

In recent years, the prevalence of MetS in children and adolescents has gradually increased throughout the world [3,4,5,6]. Sina E et al. found that the prevalence of MetS was 5.5% in European children and adolescents aged 2–16 years [7]. According to the analysis by DeBoer et al., the prevalence of MetS by U.S. census division ranged from 4.6% to 13.6% among 4600 U.S. adolescents aged 12–19 years [8]. Ye et al. conducted a meta-analysis and found that the prevalence rates of MetS were between 1.8% and 2.6% among Chinese children and adolescents aged 6–20 years [9]. These data indicate that the prevalence of MetS in children and adolescents cannot be ignored. MetS or its components in childhood and adolescence may increase the risk of chronic diseases, such as cardiovascular disease (CVD) and type 2 diabetes mellitus (T2DM) in adulthood [10,11]. The increasing prevalence of MetS in children and adolescents may have implications for future global health burdens [12].

Sedentary behavior refers to the behavior in which energy consumption does not exceed 1.5 metabolic equivalents of task (METs) when sitting or in a reclining posture in an awake state in an education, family, community, and transportation environment [13,14,15]. It includes two types of sitting: sitting without looking at the screen and sitting in front of the screen; the time of the latter is called screen time. Song et al. analyzed the surveillance data of Chinese residents’ nutrition and health status from 2010 to 2012 and found that the average daily sedentary behavior time of children aged 6–17 years was 2.92 h, and 85.8% of them had sedentary behaviors for more than two hours a day [16]. A survey conducted by Wu et al. in 12 provinces of China found that 16.2% and 41.5% of primary and middle school students, respectively, had screen time exceeding the recommended range of 2 h/day on study days and on weekends [17].

The “Guidelines on physical activity and sedentary behavior” issued by the World Health Organization (WHO) in 2020 suggests that, among children and adolescents, increased sedentary behavior is associated with the following adverse health outcomes: increased obesity, cardiovascular metabolism, and reduced sleep time; children and adolescents should limit the amount of time spent being sedentary [14]. At the same time, available studies have found that sedentary behavior is associated with an increased risk of CVD and MetS [15,18,19], and a systematic review by Tremblay et al. concluded that lower sedentary behavior, of any type, is associated with favorable health indicators [20].

China has experienced rapid social and economic changes in the past decade, consequently, lifestyle and health behaviors may have changed among Chinese children and adolescents. However, there are few descriptions of sedentary behavior in school-aged children in China, and few studies have analyzed the prevalence of MetS in school-aged children in Beijing. Therefore, we used data from the Nutrition and Health Surveillance in Schoolchildren of Beijing (NHSSB) in 2019 to describe the prevalence of MetS among children aged 6–14, analyze the relationship between sedentary behavior and MetS, and provide scientific evidence for its prevention and treatment.

## 2. Materials and Methods

### 2.1. Study Design and Participants

Beijing launched the Balanced Meal Campus Health Promotion Action Project in 2014–2020, which uses health education as the main means of guiding primary and secondary school students to develop healthy eating habits. To evaluate the changes in the nutritional health status of primary and middle school students during this period, the Beijing Center for Disease Prevention and Control conducted a baseline survey in 2015, a mid-term survey in 2017, and a third round of Nutrition and Health Surveillance in Schoolchildren of Beijing (NHSSB) in 2019. This surveillance adopted a multistage stratified cluster random sampling method (Figure 1).

This surveillance was approved by the Ethics Committee of the Beijing Center for Disease Prevention and Control (approval number: No. 14, 2019), and all participating students and their guardians signed the informed consent.

### 2.2. Questionnaires Data Collection

The questionnaires used in this study were generated after expert argumentation based on the nutrition surveillance over the years. Questionnaires were designed for students and their parents separately. Students’ nutrition, health-related knowledge, and eating behavior were obtained by the student questionnaire; basic family information, family behaviors, students’ physical activity, and health information were obtained by the parental questionnaire. Questionnaire items on sedentary behaviors include (1) watching TV; (2) playing computers, tablets, mobile phones, and other electronic devices; (3) reading newspapers, novels, and other paper reading; (4) doing homework; (5) other sedentary activities. The sedentary behavior time of children on school days and weekends was measured separately. We only counted the sedentary behavior time of children after school considering that children have almost equal sedentary time in class.

### 2.3. Anthropometric Measurements

(1) Height and weight: The measurement of height and weight was carried out in accordance with the requirements of GB/T 26343-2010 “Technical standard for physical examination for students.” Height was measured in centimeters (cm) to the nearest 0.1 cm; weight was measured in kilograms (kg) to the nearest 0.1 kg.

(2) Waist circumstance (WC): Using a tape measure, WC was measured horizontally at the midpoint between the inferior edge of the costal arch and the iliac crest in the mid-axillary line, at the end of a normal exhalation. The measurement was performed twice, with cm as the unit and accurate to 0.1 cm. The average of two repeated measurements was calculated for WC.

### 2.4. Blood Pressure Measurement

The measurement instrument was a digital sphygmomanometer (model HBP1300; Omron Healthcare (China) Co., Ltd., Liaoning, China). BP was obtained from the left arm using the appropriate cuff for each participant. The participant rested quietly for at least 5 min before the measurement and at the same time we ensured that the measurement environment was quiet and comfortable. BP was measured three times with a 1-min interval between repetitions; the average of three measures was calculated.

### 2.5. Laboratory Biochemical Examination

After 10–12 h overnight fasting, blood samples were collected by qualified medical physicians, centrifuged at 3000 rpm for 10 min within 30 min, collected the supernatant to aliquot, and then stored in a refrigerator at −80 °C. Blood glucose, blood lipids, and other biochemical indicators were tested by an automatic biochemical analyzer (model 7600; Hitachi High-Tech (China) Co., Ltd., Beijing, China) and corresponding reagents (Wako Pure Chemical Industries, Ltd., Osaka, Japan). Glucose, TG, and HDL-C were analyzed by hexokinase method, free glycerol method, and direct method, respectively.

### 2.6. Calculation or Diagnostic Criteria

#### 2.6.1. Sedentary Behavior

Sedentary behavior time was estimated by counting the minutes per day spent watching TV, computer, tablet computer, mobile phone, and other screens, reading newspapers, novels, and other paper reading time, and doing homework and other sedentary activities. Average sedentary behavior time could be calculated by the following formula:Sedentary behavior time (min/day) = (sedentary behavior time on school days × 5 + sedentary behavior time on weekends × 2)/7

The sedentary behavior time was categorized in tertiles (low, medium, and high). There were 1090, 1114, and 1075 participants in the low-level, medium-level and high-level groups, respectively.

#### 2.6.2. Metabolic Syndrome

MetS was defined according to the CHN2012 criteria [21]. Abdominal obesity is an essential component for the diagnosis of MetS. The waist-to-height ratio (WHtR) is used as the screening index [21,22,23,24], and the cut-off point of WHtR is 0.48 for boys and 0.46 for girls as the screening cut-off value for abdominal obesity [22]. At least two components of the following should be provided at the same time: (1) Hyperglycemia: a. Impaired fasting blood glucose (IFG): fasting blood glucose ≥ 5.6 mmol/L; b. or impaired glucose tolerance (IGT): oral glucose tolerance test (OGTT) 2 h blood glucose ≥ 7.8 mmol/L, but <11.1 mmol/L; c. or type 2 diabetes [21]. (2) Hypertension: using the rapid identification method, children aged ≥10 years with systolic blood pressure (SBP) ≥ 130 mmHg and/or diastolic blood pressure (DBP) ≥ 85 mmHg are identified to be hypertension; children aged 6–10 years with SBP ≥ 120 mmHg and/or DBP ≥ 80 mmHg are identified to be hypertension [22]. (3) Low HDL-C: HDL-C < 1.03 mmol/L or non-HDL-C ≥ 3.76 mmol/L. (4) High TG: TG ≥ 1.47 mmol/L [21].

### 2.7. Statistical Analysis

EpiData 3.0 software (The Epi Data Association, Odense, Denmark) was used for data entry, and all data received were double-checked. All analyses were performed with IBM SPSS version 26.0 software (IBM, Aromonk, NY, USA). Continuous variables were reported as mean ± standard deviation or the median and interquartile range (IQR), and categorical variables were reported as frequency (percentage). Chi-squared test was used to compare the constituent ratios between groups, *t*-test was used to compare the differences in height, weight, WC, glucose, and other indicators between the MetS and the non-MetS groups, nonparametric tests were used to compare differences in time to sedentary behavior between groups, and binary logistic regression was used to analyze the correlation between sedentary behavior and MetS and its components. A two-sided *p* < 0.05 was considered statistically significant.

## 3. Results

### 3.1. The Status Quo of MetS

The overall prevalence of MetS among students surveyed was 2.4%. The prevalence rates of abdominal obesity, hypertension, hyperglycemia, high TG, and low HDL-C were 23.0%, 4.0%, 7.9%, 7.8%, and 5.5%, respectively. Table 1 shows the distribution of participants’ characteristics by MetS. Boys were more likely to have MetS than girls (χ^2^ = 3.947, *p* = 0.047) and suburbs were more prevalent in the MetS group than in the non-MetS group (χ^2^ = 9.982, *p* < 0.01). The prevalence rates of MetS among students in different age groups were also different (χ^2^ = 27.463, *p* < 0.001), and the prevalence was higher with increasing age. Children in the MetS group had longer sedentary behavior time (Z = 3.409, *p* = 0.001). The high-level sedentary behavior group had more MetS than the low-level sedentary behavior group (χ^2^ = 9.363, *p* < 0.01). All components of MetS differed between the two groups, and the MetS group showed worse profiles with higher WC, WHtR, systolic and diastolic blood pressure, fasting glucose, and triglyceride, and lower HDL-C (*p* < 0.001).

### 3.2. The Status Quo of Sedentary Behavior

The median sedentary behavior time of the participants was 175.7 (128.6, 228.6) min, the mean was 182.9 ± 75.2 min, and 77.5% of participants had more than 2 h of sedentary behavior per day. There were no statistically significant differences between the sedentary behavior time of participants of different genders and regions (Z = 0.599, *p* = 0.446), but screen time of the boys was higher than that of the girls (Z = 4.034, *p* < 0.001), and screen time of suburban students was higher than that of urban students (Z = 7.697, *p* < 0.001). Older children had longer daily sedentary behavior time, and the difference in each age group was statistically significant (*p* < 0.001). Figure 2 shows the composition of sedentary behavior among children by gender, age, residence, and school days or weekends. The median sedentary behavior times on school days and weekends were 140.0 min and 240.0 min, respectively.

### 3.3. Relationship between Sedentary Behavior and MetS

Table 2 shows the prevalence of MetS components in the different sedentary behavior time groups by gender. In boys, the prevalence rates of abdominal obesity, hyperglycemia, high TG, and low HDL-C were higher in the high-level sedentary behavior group than those in the low-level sedentary behavior group (*p* < 0.05); and in girls, the prevalence rates of high TG, low HDL-C, and MetS were higher in the high-level sedentary behavior group than those in the low-level sedentary behavior group (*p* < 0.05). Figure 3 shows the percentages of people with abnormal numbers of metabolism in different sedentary behavior time groups by gender. The high-level sedentary behavior group also had more metabolic abnormalities than the low-level sedentary behavior group in both boys and girls (*p* < 0.001).

Single-factor logistic regression analysis of sedentary behavior time and MetS and its components revealed that compared with low-level sedentary behavior time, high-level sedentary behavior time had higher risks of abdominal obesity, hyperglycemia, high TG, low HDL-C, and MetS (*p* < 0.05). After adjusting for the confounding factors of age, gender, residence, caregiver’s education, per capita household income, and leisure time moderate-to-vigorous intensity physical activity (MVPA), further analysis found that, compared with children with low-level sedentary behavior, the risks of abdominal obesity, high TG, and low HDL-C were higher in children with high-level sedentary behavior (odds ratio (OR) 1.49, 95% confidence interval (CI) 1.16–1.92, *p* < 0.01; OR 1.57, 95% CI 1.05–2.35, *p* = 0.028; OR 2.02, 95% CI 1.22–2.32, *p* < 0.01, respectively) (Table 3).

A stratified analysis by gender found that compared with children with low-level sedentary behavior, the risks of abdominal obesity and low HDL-C were higher in boys with high-level sedentary behavior (OR 1.51, 95% CI 1.10–2.07, *p* = 0.011; OR 2.25, 95% CI 1.06–4.76, *p* = 0.034, respectively); the risk of abdominal obesity was higher in girls with medium and high-level sedentary behavior (OR 1.52, 95% CI 1.01–2.27, *p* = 0.043; OR 1.59, 95% CI 1.04–2.43, *p* = 0.032, respectively) (Table 4).

## 4. Discussion

This study aimed to examine the prevalence of metabolic syndrome (MetS) in children aged 6–14 years in Beijing and determine whether sedentary behavior is a risk factor. Through the cross-sectional study design, the overall prevalence of MetS among children aged 6–14 in Beijing in 2019 was 2.4%, which was consistent with the results of the 2017 Guangzhou City and 2010 nationwide surveys among the same age group [25,26], which was lower than the 3.6% in Pudong New Area, Shanghai [27]. Boys were more likely to have MetS than girls, which was consistent with the results of most studies [25,26,28]. Possible reasons for this result are the physiological structure and genetic differences between boys and girls, and the prevalence of abdominal obesity among boys is much higher than that of girls. Some studies have shown that obesity is an important factor affecting the prevalence of MetS [2,29,30]. In addition, the screen time of the boys was higher than that of the girls. Carson et al. found that current evidence suggests that screen time may have a larger impact on health than overall sedentary time [19]. Suburbs were more prevalent in the MetS group than in the non-MetS group, which may also be related to the higher obesity rate and screen time among students in suburban areas than in urban areas in Beijing [31]. The prevalence rates of MetS among students in different age groups were also different. The overall trend was that prevalence goes up with increasing age. This trend was consistent with the results of previous studies [25,28]. The reason for this result may be that individual hormone levels and body fat distribution characteristics change significantly with age [32].

The study found that the median sedentary behavior time for children aged 6–14 in Beijing was 175.7 min, and the mean was 182.9 ± 75.2 min, which was similar to the average total sedentary behavior time in Guangzhou in 2017 of 3.06 ± 1.34 h [33], and 77.5% of students had more than 2 hours of sedentary behavior per day, which was lower than the 86.2% reported by Song et al. [16]. The median screen time was 47.14 min, and the average daily screen time exceeding 2 h for children accounted for 8.2%, slightly lower than the 9.0% in Guangzhou in 2017 [25] and higher than the 6.7% in Beijing in 2018 [34]. Changes in social development and lifestyles, as well as cultural differences in different regions, may be responsible for the different sedentary behavior time and screen time of children. In addition, inconsistencies in findings may reflect inconsistencies in data collection methods. Therefore, it is necessary to establish a uniform standard for the collection of sedentary time for children to facilitate comparisons between multiple regions. In this study, almost half of the sedentary behavior time came from homework, which is a problem that should be of concern. Moreover, the government of China released the Opinions on Further Reducing the Burden of Homework and Off-Campus Training for Compulsory Education Students (“Double Reduction” policy) on 24 July 2021; future studies focused on the effect of this policy on children’s sedentary time are expected.

This study found that the prevalence of each MetS component was higher with longer sedentary time. In boys, the prevalence rates of abdominal obesity, hyperglycemia, high TG, and low HDL-C were higher in the high-level sedentary behavior group than those in the low-level sedentary behavior group; and in girls, the prevalence rates of high TG, low HDL-C, and MetS were higher in the high-level sedentary behavior group than those in the low-level sedentary behavior group. It suggested that the effects of sedentary behavior on MetS and its components might be different in different genders. Thus, gender should be considered in studies on metabolic outcomes. This study used waist-to-height ratio (WHtR) to screen for abdominal obesity. Several studies have shown that WHtR is more predictive of abdominal obesity in children and adolescents than body mass index [23,24,35]. It can be a useful substitute for body obesity when skinfold measurements are not available. Because height affects waist circumference, correcting WHtR can eliminate the effect of height and the potential impact on reference values related to age, gender, and race [36]. Research by Jorge Mota et al. found that boys with higher sedentary behaviors are more likely to suffer from central obesity [37]. Available evidence indicates that sedentary behavior is also independently associated with an increased risk of abnormal glucose metabolism, including decreased insulin sensitivity [38] and increased fasting insulin levels [39,40]. Hjorth et al. showed that a longer duration of accelerometer-derived sedentary time was significantly associated with lower high-density lipoprotein (HDL) cholesterol [41].

A longitudinal study by Greer et al. found that men with moderate (12–19 h/week) and high (>19 h/week) sedentary behaviors had 65% and 76% higher risks of MetS than men with low sedentary behaviors (<12 h/week), respectively [42]. Berg et al. used an accelerometer to measure sedentary behavior time, and further analysis found that an additional hour of static activity was associated with a 39% increase in the risk of MetS [43]. Edwardson et al. also conducted a meta-analysis and found that the longer the sedentary behavior, the greater the risk of MetS. Reducing sedentary behavior may play an important role in preventing MetS [44]. In this study, after adjusting for confounders, compared with children with low sedentary behavior time, the risk of MetS was not higher in children with high sedentary behavior time, but the risks of abdominal obesity and low HDL-C were higher in boys with high-level sedentary behavior, while the risk of abdominal obesity was higher in girls with medium and high-level sedentary behavior. A long sedentary behavior time has been suggested to be correlated with metabolic abnormalities in children. The mechanisms underlying the adverse effects of sedentary behavior on metabolic syndrome remain unclear. A possible reason for this is that prolonged sedentary behavior affects body weight and lipid metabolism [41,45].

When adjusting for confounding factors, leisure time MVPA, which might affect the prevalence of MetS, was not statistically significant, further verifying the difference between sedentary behavior and physical inactivity. A few studies have found that even if the time of MVPA meets the requirements of the recommended guidelines, excessive sedentary behavior still leads to metabolic abnormalities [42,45,46]. The amount of physical activity does not cancel out the negative effects of sedentary behavior, so it should be clear that sedentary behavior and physical inactivity are two different concepts that affect health independently.

However, this study has some limitations. First, this study was a cross-sectional investigation, which could not clarify the causal relationship between sedentary behavior and MetS. Therefore, a prospective study should be conducted. Second, the collection of sedentary behavior time depended on students’ self-reports or parents’ reports, which were prone to recall and reporting bias. In addition, sedentary traffic time refers to the behavior time in which energy consumption does not exceed 1.5 METs in the transportation environment. However, due to the lack of a feasible and effective measurement method for sedentary traffic time, it is not involved in this paper, which may lead to a lower total sedentary time and underestimate its influence. Therefore, objective devices such as accelerometers, inclinometers, and sports bracelets should be used to measure the level of sedentary behavior, or a reliable and effective subjective measurement method of sedentary behavior should be developed and used regularly. Third, the International Diabetes Federation (IDF) considers impaired fasting glucose (fasting blood glucose (FPG) ≥5.6 mmol/L) as an independent indicator of abnormal glucose metabolism. However, clinical studies have found that FPG is not stable enough for being easily influenced by factors such as emotion, stress, and diseases [22]. Therefore, OGTT or repeated measurement of FPG should be performed in future experiments.

## 5. Conclusions

Higher sedentary behavior time is related to the higher risk of MetS components among children aged 6–14 years in Beijing. Reducing sedentary behavior may be an important method for preventing metabolic diseases.

## Figures and Tables

**Figure 1 nutrients-14-01869-f001:**
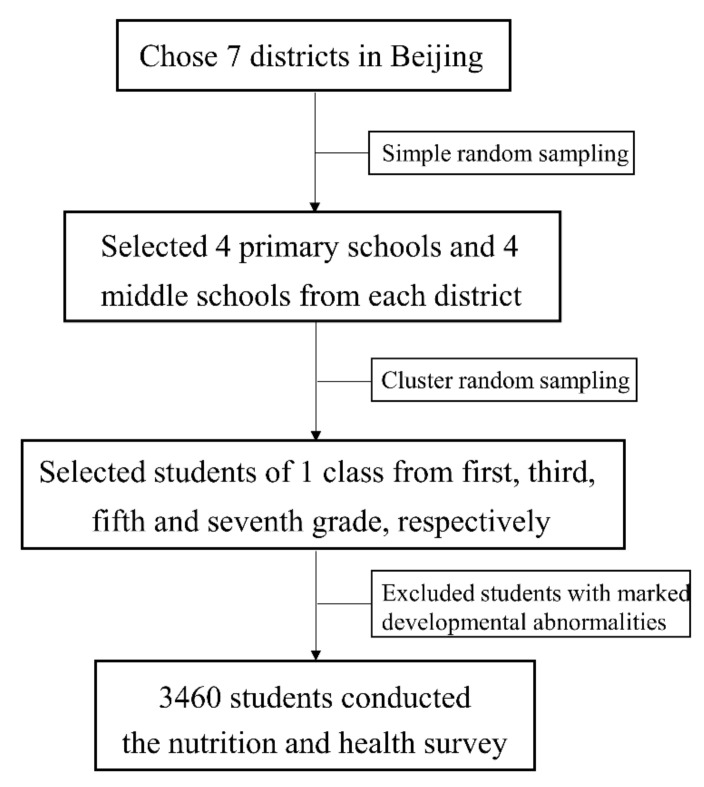
Flow diagram of the study participants.

**Figure 2 nutrients-14-01869-f002:**
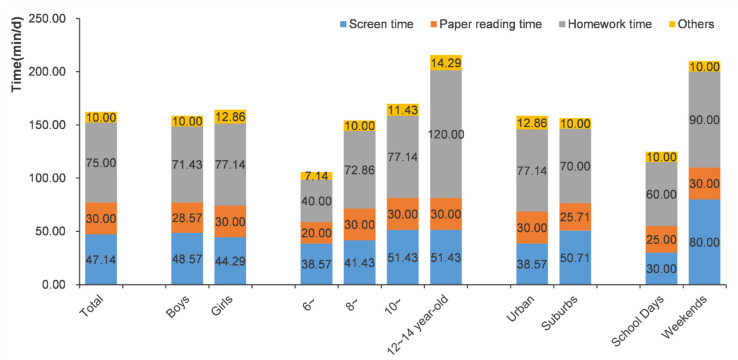
The composition of sedentary behavior time among children by gender, age, residence, and on school days or weekends.

**Figure 3 nutrients-14-01869-f003:**
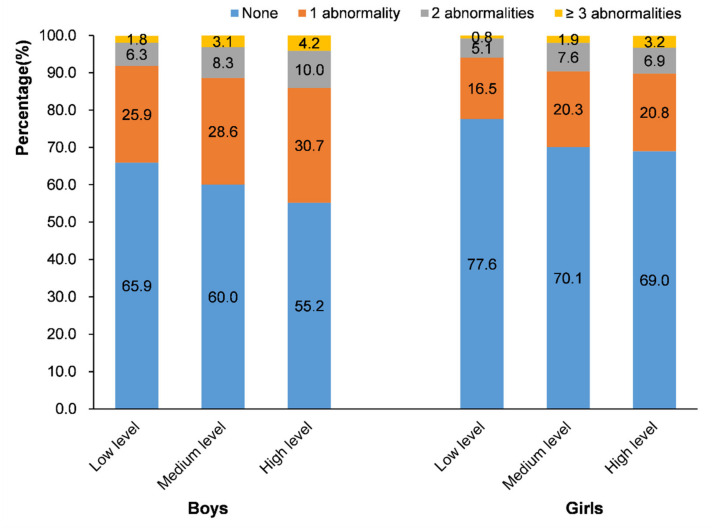
Percentages of people with abnormal numbers of metabolism in different sedentary behavior time groups by gender.

**Table 1 nutrients-14-01869-t001:** Characteristics of participants by MetS #.

Group	Total (*n* = 3392)	Non-MetS (*n* = 3311)	MetS (*n* = 81)	*p*-Value
Gender				0.047 *
Boy	1724(50.8)	1674(50.6)	50(61.7)	
Girl	1668(49.2)	1637(49.4)	31(38.3)	
Age (years)				<0.001 ***
6~8	895(26.4)	886(26.8)	9(11.1)	
8~10	805(23.7)	796(24.0)	9(11.1)	
10~12	859(25.3)	831(25.1)	28(34.6)	
12~14	833(24.6)	798(24.1)	35(43.2)	
Residence				0.002 **
Urban	1328(39.2)	1310(39.6)	18(22.2)	
Suburbs	2064(60.8)	2001(60.4)	63(77.8)	
Height (cm)	140.95 ± 15.21	140.67 ± 15.13	152.33 ± 14.53	<0.001 ***
Weight (kg)	37.52 ± 14.12	36.96 ± 13.59	60.84 ± 15.86	<0.001 ***
Average sedentary time (min/day)	175.7(128.6228.6)	175.1(127.1227.9)	208.9(154.6251.8)	0.001 **
Sedentary behavior time group				0.009 **
Low level	1069(33.3)	1055(33.6)	14(18.4)	
Medium level	1091(33.9)	1064(33.9)	27(35.5)	
High level	1055(32.8)	1020(32.5)	35(46.1)	
Caregiver’s education				0.125
Junior high school or below	477(14.1)	462(14.0)	15(18.5)	
High school/Technical school	640(18.9)	621(18.8)	19(23.5)	
College/Vocational college	703(20.7)	683(20.6)	20(24.7)	
Undergraduate or above	1571(46.3)	1544(46.6)	27(33.3)	
Per capita household income (CNY/year)				0.257
<20,000	307(9.1)	297(9.0)	10(12.3)	
20,000~39,999	489(14.4)	473(14.3)	16(19.8)	
40,000~69,999	721(21.3)	702(21.2)	19(23.5)	
≥70,000	1427(42.1)	1402(42.3)	25(30.9)	
Not clear	448(13.2)	437(13.2)	11(13.6)	
Leisure time MVPA (min/week)				0.826
≤60	1151(40.3)	1126(40.4)	25(39.7)	
61~120	710(24.9)	693(24.8)	17(27.0)	
121~240	611(21.4)	600(21.5)	11(17.5)	
>240	381(13.4)	371(13.3)	10(15.9)	
WC (cm)	62.40 ± 11.37	61.71 ± 10.45	82.60 ± 8.61	<0.001 ***
WHtR	0.44 ± 0.06	0.44 ± 0.06	0.54 ± 0.05	<0.001 ***
SBP (mmHg)	109.03 ± 10.14	108.62 ± 9.73	121.26 ± 11.16	<0.001 ***
DBP (mmHg)	64.32 ± 7.08	64.18 ± 7.01	68.12 ± 7.31	<0.001 ***
Fasting glucose (mmol/L)	5.03 ± 0.43	5.03 ± 0.42	5.36 ± 0.51	<0.001 ***
Serum TG (mmol/L)	0.78 ± 0.47	0.75 ± 0.41	1.83 ± 0.92	<0.001 ***
Serum HDL-C (mmol/L)	1.46 ± 0.29	1.47 ± 0.29	1.01 ± 0.18	<0.001 ***

#: Data were presented as means ± standard deviations, or median (P_25_, P_75_), or frequency values (percentages, %). Abbreviations: WC, waist circumference; WHtR, waist-to-height ratio; SBP, systolic blood pressure; DBP, diastolic blood pressure; TG, triglyceride; HDL-C, high-density lipoprotein cholesterol; MVPA, moderate-to-vigorous intensity physical activity. *: *p* < 0.05, **: *p* < 0.01, ***: *p* < 0.001.

**Table 2 nutrients-14-01869-t002:** Prevalence of MetS components in different sedentary behavior time groups by gender #.

	Boys				Girls			
	Low Level	Medium Level	High Level	*p* Value	Low Level	Medium Level	High Level	*p* Value
Abdominal obesity	140(25.2)	168(30.3)	172(33.0)	0.018 *	68(13.2)	91(17.0)	90(16.9)	0.161
Hypertension	22(3.9)	25(4.4)	35(6.5)	0.101	13(2.5)	21(3.8)	17(3.2)	0.460
Hyperglycemia	38(6.7)	50(8.8)	61(11.4)	0.024 *	25(4.8)	33(6.0)	42(7.8)	0.123
High TG	30(5.3)	43(7.6)	50(9.3)	0.036 *	29(5.5)	50(9.1)	54(10.0)	0.020 *
Low HDL-C	16(2.8)	29(5.1)	38(7.1)	0.005 **	19(3.6)	37(6.7)	41(7.6)	0.017 *
MetS	10(1.8)	17(3.1)	19(3.6)	0.174	4(0.8)	10(1.9)	16(3.0)	0.031 *

#: Data were presented as frequency values (percentages, %). *p*-values were obtained by the chi-squared test, *: *p* < 0.05, **: *p* < 0.01. Abbreviations: TG, triglyceride; HDL-C, high-density lipoprotein cholesterol; MetS: metabolic syndrome.

**Table 3 nutrients-14-01869-t003:** Logistic regression analysis of sedentary behavior time, MetS, and its components of children aged 6–14 years (OR (95% CI of OR)).

Model	Sedentary Time	Abdominal Obesity	Hypertension	Hyperglycemia	High TG	Low HDL-C	MetS
Crude	Low level	1 (ref)	1 (ref)	1 (ref)	1 (ref)	1 (ref)	1 (ref)
	Medium level	1.29(1.05,1.59) *	1.30(0.83,2.03)	1.31(0.94,1.84)	1.59(1.14,2.23) **	1.90(1.25,2.89) **	1.91(1.00,3.67)
	High level	1.37(1.11,1.68) **	1.53(0.99,2.37)	1.73(1.25,2.39)**	1.87(1.34,2.61) ***	2.39(1.59,3.59) ***	2.59(1.38,4.83) **
Model 1	Low level	1 (ref)	1 (ref)	1 (ref)	1 (ref)	1 (ref)	1 (ref)
	Medium level	1.25(0.98,1.59)	1.05(0.61,1.82)	1.01(0.67,1.51)	1.44(0.98,2.14)	1.57(0.95,2.58)	1.39(0.65,2.96)
	High level	1.49(1.16,1.92) **	1.23(0.71,2.14)	1.26(0.84,1.89)	1.57(1.05,2.35) *	2.02(1.22,3.32) **	1.63(0.76,3.47)

Crude: Unadjusted confounding variables; Model 1: Adjusted for age, gender, residence, caregiver’s education, per capita household income, and leisure time moderate-to-vigorous intensity physical activity; ref: reference; *: *p* < 0.05, **: *p* < 0.01, ***: *p* < 0.001.

**Table 4 nutrients-14-01869-t004:** Logistic regression analysis of sedentary behavior time, MetS, and its components of children aged 6–14 years by gender # (OR (95% CI of OR)).

Sedentary Time	Abdominal Obesity	Hypertension	Hyperglycemia	High TG	Low HDL-C	MetS
Boys						
Low level	1 (ref)	1 (ref)	1 (ref)	1 (ref)	1 (ref)	1 (ref)
Medium level	1.15(0.85,1.56)	0.67(0.32,1.41)	1.06(0.62,1.81)	1.35(0.77,2.36)	1.53(0.71,3.30)	1.18(0.45,3.06)
High level	1.51(1.10,2.07) *	0.91(0.44,1.86)	1.41(0.83,2.40)	1.53(0.87,2.70)	2.25(1.06,4.76) *	1.43(0.56,3.68)
Girls						
Low level	1 (ref)	1 (ref)	1 (ref)	1 (ref)	1 (ref)	1 (ref)
Medium level	1.52(1.01,2.27) *	1.87(0.80,4.37)	0.89(0.47,1.68)	1.56(0.89,2.71)	1.61(0.83,3.12)	2.08(0.58,7.42)
High level	1.59(1.04,2.43) *	1.89(0.77,4.64)	1.03(0.54,1.96)	1.61(0.91,2.86)	1.83(0.93,3.59)	2.25(0.61,8.26)

#: Adjusted for age, residence, caregiver’s education, per capita household income, and leisure time moderate-to-vigorous intensity physical activity; *: *p* < 0.05.

## Data Availability

The data presented in this study are available on request from the corresponding author. The data are not publicly available due to privacy.

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
