# Peer review of "Self-Reported Sedentary Behavior and Metabolic Syndrome among Children Aged 6–14 Years in Beijing, China"

_nutrients, 2022, doi:10.3390/nu14091869_

Round 1
Reviewer 1 Report
The authors conducted a cross-sectional analysis of data from Nutrition and Health Surveillance in Schoolchildren of Beijing, to investigate associations between sedentary time and presence of MetS, and risk factors which comprise MetS. Overall, the study was well executed and the results appropriately interpreted. The manuscript is well written.
TITLE
Suggest that the title of the manuscript be edited to "Self-reported sedentary..."
METHODS
Given that sedentary time is the predictor variable, more information about the questionnaire used to collect these data is needed. Were only two questions about sedentary behaviour asked? That is, average number of hours per day spent in sedentary activities during the week and on weekends? Were these open-ended questions where parents simply entered a best guess as to their children's sedentary time? How would the parents be able to determine this during the week, when children are in school? Elaboration is needed.
When was WC measure relative to respiration? Was it mid-respiration (after exhalation, before inhalation)?
For how long had children fasted prior to phlebotomy?
RESULTS
Table 1: Please present the age ranges rather than "~6", "~8" etc., as you did for MVPA in Table 1.
DISCUSSION
How do the authors explain the increasing prevalence of MetS with increasing caregiver education and household income?
Lines 263-264: the point that inconsistencies in findings may reflect inconsistencies in data collection methods would benefit from elaboration.
Lines 301-304: This is an excellent point.
Line 309: What is "static traffic time"?
Lines 313-316: "Third, although the International Diabetes Federation (IDF) considers impaired fasting glucose (IFG), that is, fasting blood glucose (FPG) ≥5.6 mmol/L, as an independent indicator of abnormal glucose metabolism, it has been widely recognized." What does this mean?
WRITING
There are minor issues with English throughout. Some illustrative examples:
- Line 112: "a uniform soft ruler". What does uniform mean in this context? I believe that the authors mean a tape measure vs. soft ruler.
- Line 118: "The measuring instrument used..." --> "The measurement instrument was..."
- Lines 250-251: "the trend of high and rising" --> please clarify
Author Response
Dear Reviewer,
Thank you for your comments concerning our manuscript (ID: nutrients-1704731). Those comments are all valuable and very helpful in revising and improving our manuscript, as well as providing important guidance to our contemporary and further researches. Thus, we have discussed comments carefully and made correction. Revised portions are marked up using the “Track Changes” function in the manuscript. The main corrections in the paper and response to your comments are as following:
Q1.Suggest that the title of the manuscript be edited to "Self-reported sedentary..."
A1: Thank you for your helpful suggestion. We have modified our title in the manuscript:
“Self-reported sedentary behavior and metabolic syndrome among children aged 6-14 years in Beijing, China”
Q2.Given that sedentary time is the predictor variable, more information about the questionnaire used to collect these data is needed. Were only two questions about sedentary behaviour asked? That is, average number of hours per day spent in sedentary activities during the week and on weekends? Were these open-ended questions where parents simply entered a best guess as to their children's sedentary time? How would the parents be able to determine this during the week, when children are in school? Elaboration is needed.
A2: We are sorry for our inexplicit demonstration in the manuscript. Actually, we asked five questions about sedentary behaviors and average time per day spent in sedentary activities during the week and on weekends. We have revised our expression in methods section(Lines 113-116):
“Questionnaire items on sedentary behaviors include: 1) watching TV; 2) playing computers, tablets, mobile phones, and other electronic devices; 3) reading newspapers, novels, and other paper reading; 4)doing homework; 5) other sedentary activities. The sedentary behavior time of children on school days and weekends was measured separately.”
Secondly, we only counted the sedentary behavior time of children after school considering that children have almost equal sedentary time in class. Therefore, parents were able to determine the children’s sedentary time. We have added it in the methods section(Line 117):
“We only counted the sedentary behavior time of children after school considering that children have almost equal sedentary time in class.”
Sedentary time was estimated by parents or children, which may lead to reporting bias. We have mentioned it in limitation section(Lines 314-315):
“The collection of sedentary behavior time depended on students' self-reports or parents' reports, which were prone to recall and reporting bias.”
Q3.When was WC measure relative to respiration? Was it mid-respiration (after exhalation, before inhalation)? Line 112: "a uniform soft ruler". What does uniform mean in this context? I believe that the authors mean a tape measure vs. soft ruler.
A3: Thank you for your helpful comment. We have added and revised it in the methods section(Line 128-132):
“Using a tape measure, WC was measured horizontally at the midpoint between the inferior edge of the costal arch and the iliac crest in the mid-axillary line, at the end of a normal exhalation.”
Q4.For how long had children fasted prior to phlebotomy?
A4: Blood samples were collected after a 10-12h overnight fasting. We have corrected “participants fasted overnight” to “a 10-12h overnight fasting”(Line 144).
Q5.Table 1: Please present the age ranges rather than "~6", "~8" etc., as you did for MVPA in Table 1.
A5: Thank you for your suggestion. We have revised it in the manuscript.
Q6.How do the authors explain the increasing prevalence of MetS with increasing caregiver education and household income?
A6: As shown in the manuscript, we have compared the differences between groups and found no statistical significance. The number of caregivers with/above bachelor's degree accounts for the highest percentage in our research, leading to a similar phenomenon in proportion of children with MetS. In fact, the prevalence rates of MetS in Junior high school or below, High school/Technical school, College/Vocational College, Undergraduate or above are 3.1%, 3.0%, 2.8%, and 1.7%, respectively. Similarly, for household income, the prevalence rates of MetS in <20,000, 20000~39999, 40000~69999, ≥70,000, and Not clear group are 3.3%, 3.3%, 2.6%, 1.8%, and 2.5%, respectively. The prevalence of MetS seems to be lower with increasing caregiver’s education and household income, but there is no statistically significant.
Q7.Lines 263-264: the point that inconsistencies in findings may reflect inconsistencies in data collection methods would benefit from elaboration.
A7: Thank you for your suggestion. We have revised it in the manuscript.
Q8.Line 309: What is "static traffic time"?
A8: We’re sorry for not clarifying the statement of “static traffic time”. Static traffic time refers to the behavior time in which energy consumption does not exceed 1.5 METs in the transportation environment. We have added and revised it in the discussion section(Lines 364-367):
“Sedentary traffic time refers to the behavior time in which energy consumption does not exceed 1.5 METs in the transportation environment. However, due to the lack of a feasible and effective measurement method for sedentary traffic time, it is not involved in this paper, which may lead to a lower total sedentary time and underestimate its influence.”
Q9.Lines 313-316: "Third, although the International Diabetes Federation (IDF) considers impaired fasting glucose (IFG), that is, fasting blood glucose (FPG) ≥5.6 mmol/L, as an independent indicator of abnormal glucose metabolism, it has been widely recognized." What does this mean?
A9: We’re sorry for the inappropriate expression. Actually, impaired fasting glucose is widely used as an independent indicator of abnormal glucose metabolism according to the International Diabetes Federation (IDF). We have rewritten this sentence in the discussion section as following(Lines 372-374):
"Third, the International Diabetes Federation (IDF) considers impaired fasting glucose (fasting blood glucose (FPG) ≥5.6 mmol/L) as an independent indicator of abnormal glucose metabolism. However, clinical studies have found that FPG is not stable enough for being easily influenced by factors such as emotion, stress, and diseases[22]. Therefore, OGTT or repeated measurement of FPG should be performed in future experiments."
Q10.Line 118: "The measuring instrument used..." --> "The measurement instrument was..."
A10: Thank you for your suggestion. We have replaced it in the manuscript.
Q11.Lines 250-251: "the trend of high and rising" --> please clarify
A11: Thank you for your comment. “The trend of high and rising” means “The overall trend was that prevalence goes up with increasing age”. We have revised it in discussion section(Line 290):
“The prevalence rates of MetS among students in different age groups were also different. The overall trend was that prevalence goes up with increasing age. This trend was consistent with the results of previous studies.”
We tried our best to improve the manuscript and made some changes in the manuscript. These changes will not influence the content and framework of the paper. And here we did not list the changes but marked up using the “Track Changes” function in the revised paper. We appreciate for your warm work earnestly, and hope that the correction will meet with approval. Once again, thank you very much for your comments and suggestions.
Kind regards,
Junbo Wang

Reviewer 2 Report
I had a great privilege to review the manuscript entitled " Relationship between sedentary behavior and metabolic syn-2 drome among children aged 6-14 years in Beijing, China" submitted by the authors. The study described the prevalence of MetS among 78 children aged 6-14, and evaluated the relationship between sedentary behavior and MetS.
I only have several comments:
- line 30, 31-32 It is better to replace the “increase” with “ higher” in the text. The “increase” implies a potential causal relationship, whereas this is a observational study.
- Methods: Please provide more information on the sedentary behavior time during the weekend. What’s meaning of “ During weekends and holidays, the average daily 105 time of the above behaviors of children”? Line 105.
- Results:
- Table 1: Please also provide the specific sedentary behavior time.
- Have you evaluated the interaction between sex and sedentary behavior time on the risk of MetS? A stratified analysis by sex is needed.
- Discussion:
I noticed that almost half of the sedentary behavior time is come from homework, which is matter of concern. Future policy development and intervention trials should focus on this phenomenon.
